# Analysis of Teacher Self-Efficacy and Its Impact on Sustainable Well-Being at Work

**DOI:** 10.3390/bs14070563

**Published:** 2024-07-04

**Authors:** Mercedes Arias-Pastor, Steven Van Vaerenbergh, Jerónimo J. González-Bernal, Josefa González-Santos

**Affiliations:** 1Department of Education, University of Cantabria, 39005 Santander, Spain; mercedes.ariaspastor@unican.es; 2Department of Mathematics, Statistics and Computing, University of Cantabria, 39005 Santander, Spain; 3Department of Health Sciences, University of Burgos, 09001 Burgos, Spain; jejavier@ubu.es (J.J.G.-B.); mjgonzalez@ubu.es (J.G.-S.)

**Keywords:** teacher self-efficacy, sustainable well-being, initial training, inclusive education, secondary education, diversity

## Abstract

This study evaluates teacher self-efficacy perceptions among students in the Master’s Degree in Secondary Education and Baccalaureate, Vocational Training, and Language Teaching (MDSE), as well as the variables influencing these perceptions and their connection to the program’s training. The research sheds light on how self-efficacy affects views on concerns, feelings, and attitudes towards diversity and inclusive education in the current educational landscape. Out of 205 female and 100 male MDSE students surveyed, who are either graduates or nearing completion, data were gathered using the “Teacher Education in Secondary Education: Key Elements for Teaching in an Inclusive School for All” (FORPES-IN) questionnaire distributed across Spanish universities. Three primary instruments from the questionnaire were utilized: the Teachers’ Self-Efficacy Short Form (TSES-SF), the Questionnaire for Future Secondary Education Teachers regarding Perceptions of Diversity, and the Revised Scale of Feelings, Attitudes, and Concerns about Inclusive Education (SACIE-R). Findings suggest that the majority of prospective teachers exhibit moderate-to-high levels of self-efficacy. Variables such as non-formal teaching experiences, the reason for joining the MDSE program, and regular interactions with vulnerable individuals, especially in Social and Health Science domains, moderately influence self-efficacy. This study reveals a strong link between the received training and the perceived level of self-efficacy. In particular, participants with higher self-efficacy feel better equipped to handle classroom diversity and rate the MDSE program positively. Areas for enhancement are identified, such as classroom management and diversified assessment strategies. Finally, a positive correlation is observed between high self-efficacy and positive attitudes toward disability, inclusive education principles, and reduced apprehensions about inclusive teaching.

## 1. Introduction

The concept of teacher self-efficacy plays a pivotal role in shaping educators’ approaches to inclusive education. A recent study focused on students from the Master’s Degree in Secondary Education and Baccalaureate, Vocational Training, and Language Teaching (MDSE), evaluating these students’ perceptions of diversity and the training they received to become inclusive educators. This research revealed that perceived teacher self-efficacy was instrumental in promoting inclusive school principles and values and acted as a predictor for heightened readiness in diversity awareness. In addition, once a high level of teacher self-efficacy is achieved, the motivation to teach will be greater, as will their effectiveness in implementing inclusive practices, directly impacting the quality of their teaching [1,2] and student well-being [3].

Furthermore, as highlighted by Bueno-Alvarez et al. in their review on the subject [1], several authors have expressed interest in the relevance of the variables that mediate between teaching motivation and student outcomes, with the sense of self-efficacy being one of the most significant [4].

Rooted in Albert Bandura’s social cognitive theory [5], the perception of self-efficacy emerges as a crucial determinant in task and goal accomplishment. This perception is influenced by individuals’ thoughts and beliefs about their ability to plan and carry out the necessary actions to achieve desired outcomes. Bandura further states that “if people believe they have no power to produce these outcomes, they will not make the effort to make it happen”. Thus, teacher self-efficacy is defined as “teachers’ belief in their ability to organize and execute the actions necessary to successfully perform a specific teaching task in a given context” [6]. This is considered a predictor of teachers’ future behavior in terms of the effort and persistence they will dedicate to teaching, and their commitment to supporting the learning of all students based on their optimism and motivation [7]. As we can see, self-efficacy beliefs exert influence over behavior as a whole through cognitive, motivational, emotional, and selective processes.

A key study in the field [8] suggests that educators who possess strong self-efficacy beliefs tend to embrace novel concepts, willingly experiment with new teaching methods, adeptly structure their lessons, and exhibit heightened enthusiasm in their teaching approach. Furthermore, teacher self-efficacy is not exclusively connected to the academic achievements of students; it also corresponds with the level of motivation teachers exhibit during their classes and the standards they establish for their students [9].

Similarly, teachers who feel unable to motivate students, improve their own teaching, and manage the classroom often face increased lack of motivation and disillusionment in their work. These beliefs hold importance not just at opposing ends but also at moderate levels of self-efficacy, as evidenced in other studies [10], which found significant differences in motivation between teachers with intermediate levels of self-efficacy and those characterized by high levels of self-efficacy.

The prevailing model of teacher self-efficacy [11] delineates two interconnected dimensions that shape this judgment. Firstly, teachers evaluate the obstacles and enablers they encounter in the teaching and learning process. Secondly, they evaluate their own teaching competence, skills, knowledge, etc., to operate effectively in a specific teaching context. The interplay between these dimensions culminates in a self-perception of efficacy that frames the educational context as either a challenge or a threat for future teachers.

And what task currently lies at the heart of the challenges for future secondary education teachers? Within the current global framework of education and the regulatory developments in our country [12,13], two main priorities stand out: inclusive education and sustainable development [14]. There is a pressing need for quality attention to diversity in classrooms [15]. Moreover, we must shift our perspective towards finding new responses and strategies to ensure that everyone has access, participates, and achieves success within the same social and educational setting in schools where the sustainability of educational and social change is possible.

A school that promotes the well-being not only of students but also of teachers from an ecological perspective, where both are interconnected, should not cause the former to have a negative environmental impact on the latter from a perspective of sustainable well-being. Therefore, ensuring a sense of self-efficacy that promotes sustainable well-being in teachers not only benefits educators individually but also has a positive impact on the educational environment as a whole, improving the quality of teaching and learning and contributing to the academic and personal success of students [16].

The challenge of training inclusive teachers is significant and cannot be overlooked. Each teacher’s judgments about their abilities and competencies are crucial for their development in this role [17]. These elements are key to effectively implementing our educational principles and goals [18]. Furthermore, these perceptions affect the overall classroom climate and student learning outcomes [19].

Furthermore, there seems to be agreement in identifying these beliefs of teacher self-efficacy as a relevant element that influences the strategies teachers implement in their classrooms [20]; that is, how teachers respond to the diversity of situations in the classroom will be associated with the assessment of their own strengths. However, for effective teaching practice [21], it is not enough to have an adequate vision of one’s own capacity [22]; it also requires knowledge of the subject matter to be taught and mastery of a series of competencies and skills, including those outlined in the Profile of the Inclusive Teacher [17].

Therefore, it is evident that the ideas and principles of inclusive education and sustainable development require future teachers who can put them into practice in their day-to-day work in the classroom. However, this also requires them to have initial and adequate training that aligns with the challenge [20,22]. Training should empower them to develop the necessary skills and competencies. Additionally, systematic educational policies should be promoted [23] to overcome obstacles and enhance the facilitators of the socio-educational process. Positive self-efficacy beliefs are especially significant as they not only predict learning outcomes but also foster teacher motivation and commitment to their school and teaching [24]. 

Several authors [25] point out that if teachers do not feel prepared to work with all students, the challenge is to improve the training they receive by adapting it to their teaching needs and strengthening the sense of self-efficacy. Otherwise, a negative sense of self-efficacy could generate unfavorable attitudes towards the challenge posed by inclusive education and attention to diversity [26], along with higher levels of concern and stress in the daily experience of the classroom [27], lower levels of job satisfaction [28], and motivational deficits [29]. This, in turn, would not only result in the failure of process implementation but also in negative levels of teacher well-being and its associated consequences [30,31], both for them and for the entire system from a systemic perspective [32]. 

At this point, the training received in the Master’s Degree in Secondary Education and Baccalaureate, Vocational Training, and Language Teaching (MDSE) will be of great importance, not only because it is essential for the future development of education [33], but also because the level of preparation and academic training plays a crucial role in the development of self-efficacy: having adequate training, both methodological and academic, fosters positive beliefs about one’s capacity and ability to handle the required teaching actions [34], which is associated with higher self-efficacy for teaching and motivation to teach [6,35]. 

And, as José Antonio Marina Torres pointed out in the foreword of the TALIS 2013 Secondary Report [36], the quality of an educational system cannot be higher than the quality of its teachers. Therefore, if we want to combine sustainable well-being [16], quality, equity, and inclusion for all students, especially the most vulnerable ones, we must focus on teachers starting from their initial and ongoing training. And considering the significance of motivation and self-efficacy in teachers’ work, we inquire whether the training they receive in MUPES, as a precursor to professional practice, enables them to reach a high level of self-efficacy.

Given the expanded initial study sample and for the reasons previously outlined, this study aims to assess the level of perceived teacher self-efficacy among students in the MDSE program, both globally and in each of the areas that make up this construct. Additionally, the study seeks to examine the variables that modulate this perception in order to boost them, the relationship between teacher self-efficacy and the training received in the master’s program, and its ultimate impact on concerns related to the development of inclusive education as a factor that affects well-being. This includes the appropriateness of their feelings towards people with disabilities and educational needs and on attitudes to face the current educational landscape. We hypothesize that a heightened sense of teacher self-efficacy will be related to a more favorable disposition towards diversity, fewer related concerns, and a higher level of knowledge and competencies acquired in the MDSE program for their development as inclusive teachers [12] in a sustainable school for all that ensures not only the right to education in terms of quality and equity but also contributes to the shaping of just and sustainable societies [37].

## 2. Materials and Methods

### 2.1. Participants and Procedure

The sample consisted of 305 students from 47 Spanish universities who completed their studies in the MDSE program during the 2021–2022 and 2022–2023 academic years. The only requirement for inclusion in the study was to have completed the generic module of the master’s program and to have completed the internships in secondary education centers.

The FORPES-IN questionnaire was made available for online, anonymous, and voluntary completion. The questionnaire was distributed through the Google Forms platform, and universities were contacted to request their collaboration in disseminating the questionnaire to their students. Email communications were sent to the MDSE units, including a letter for the students with a link to the questionnaire.

Once the data were collected (see Appendix A), a matrix was created for evaluation using the statistical software IBM SPSS (Statistical Package for the Social Sciences) version 25.

### 2.2. Assessments

To collect the information, a 99-item questionnaire called “Teacher Training in Secondary Education: Key Elements for Teaching in an Inclusive School (FORPES-IN)” was used. This questionnaire includes five validated instruments with Spanish samples that are relevant to the study and have been used in other related research within a broader project [38]. For this specific investigation, the following instruments were selected:

To measure the sense of teacher self-efficacy, the Short Scale of Teacher Self-Efficacy [39] was used. This is the Spanish adaptation of the Teachers’ Sense of Efficacy Scale [8]. This instrument measures the perception of teacher self-efficacy in three dimensions: (a) perceived efficacy to optimize instruction, (b) perceived efficacy to manage the classroom, and (c) efficacy to engage students in learning. This scale has demonstrated high reliability and excellent validity [40], showing adequate psychometric properties in Spanish samples. The level of teacher self-efficacy is assessed on a Likert scale ranging from 1 to 9 points (not capable at all–completely capable) with anchors at 1—nothing, 3—very little, 5—some influence, 7—quite a bit, and 9—a great deal [8].

Two questionnaires were employed to assess the level of competencies, skills, and abilities acquired in the MDSE program for professional development, as well as attitudes, concerns, and feelings towards inclusive education These questionnaires are rated on a four-point Likert scale: 1 (totally disagree) to 4 (totally agree). They are the “Questionnaire for Future Secondary Education Teachers Regarding Perceptions of Diversity Attention (CFDPAD)” [41] and the Scale of Feelings, Attitudes, and Concerns about Inclusive Education, Revised (SACIE-R) in its Spanish version [42].

The CFDPAD gathers information through 43 items with high reliability (Cronbach’s Alpha) regarding: Factor 1: determining elements of the diversity attention process in the classroom (α = 0.959); Factor 2: curricular and organizational response to diversity in the classroom (α = 0.915); Factor 3: teacher training towards diversity (α = 0.870); Factor 4: formative teaching practice in diversity attention (α = 0.906); and Factor 5: teacher perception towards students with specific educational support needs (α = 0.916).

The SACIE-R, developed by Forlin et al. [43], is designed for both practicing and pre-service teachers and consists of 12 items that assess the perception of inclusive education and the concept of students who are included in it (attitudes), feelings towards people with disabilities, and concerns about having diverse students in the classroom. In the Spanish version, its reliability was found to be acceptable for pre-service teachers according to Cronbach’s Alpha (α = 0.67), which is similar to the original version (α = 0.74).

Lastly, the comprehensive instrument comprised an introductory section aimed at collecting sociodemographic information, along with data on other pertinent variables such as regular close interactions with vulnerable individuals under diverse parameters, motivations for pursuing the MDSE, etc. Furthermore, in the concluding segment of the instrument, participants were afforded the opportunity to offer open-ended comments on the questionnaire and the discussed topics to express their viewpoints. 

### 2.3. Statistical Analysis

In addition to descriptive analysis of the obtained responses, the following statistical methods were used for data analysis [44]. Initially, a bivariate analysis was performed utilizing the T-test to assess variances in the central tendencies of questionnaire responses when comparing two distinct groups. This analysis was applied to variables such as gender (female/male), regular close contact with individuals in situations of special vulnerability (yes/no), teaching experience with individuals in situations of special vulnerability in non-formal contexts (yes/no), and the type of institution where the MDSE is pursued (public/private). Secondly, ANOVA tests were used to compare differences in central tendency when the comparison criterion involved more than two groups, and post hoc DMS analyses were conducted to examine the nature of statistically significant differences between groups once identified.

A correlation analysis (Pearson) was also performed between the dimensions of the instruments used to determine the possible existence of significant and reciprocal relationships among the studied variables and, for some data, cross-tabulation with a chi-squared test is used to analyze the interaction between two variables for the identification of trends and correlations among the parameters. Subsequently, the assessment of corrected standardized residuals proves to be the most effective tool available for accurately interpreting the significance of the relationships [45].

It should be noted that, for handling the results of the “Teachers’ Sense of Efficacy Scale-Short Form (TSES-SF)”, we used the total score as a more suitable indicator for future teachers [46], subsequently examining each of the components and providing relevant data that yielded significance. Understanding that the sample has already completed the entire training process, including the practices in centers, and is not at the beginning of the training, the assessment of the three dimensions is considered adequate [47]. The participants’ response to the initial question in every item “How much can you do to” was recorded with a nine-point scale for each item, with anchors at 1: nothing, 3: very little, 5: some influence, 7: quite a bit, and 9: a great deal. For cases where the variable was identified as significant for the study objective, the scores of each subject were classified as “low”, “medium” or “high” by calculating the 33rd and 66th percentiles for this sample.

In all the conducted analyses, the significance level used was α = 0.05. All reported results have been shown to be statistically significant.

## 3. Results

A total of 305 individuals (100 male students and 205 female students) who have completed their master’s studies or are in the final stages of their studies participated in the study. They have already completed the generic module and conducted their internships in secondary education centers. The sample consists of 67.2% female and 32.8% male participants. Nearly half of the sample is in the group of “Older than 31 years old” (47.9%). Regarding the ownership of the universities, the sample is more representative of private universities (52.1%) than public universities (47.9%). It should be noted that 63% of the sample does not have regular and close contact with individuals in situations of special vulnerability, and 69.5% do not have teaching experience with vulnerable individuals in non-formal contexts. Finally, in terms of the specialties in which the MDSE students are enrolled, the most represented are Foreign Languages (14.8%), Biology and Geology (13.5%), Geography and History (11.8%), Language and Literature (9.8%), Training and Career Counseling (7.2%) and Educational Guidance and Technology (6%) (Table 1).

The following are the results regarding the perception of MDSE students regarding their sense of teacher self-efficacy, their strengths and weaknesses within the explored areas of competence, and their relationship with the training received in the MDSE. Subsequently, the relationship between the perceived level of teacher self-efficacy and the data obtained on the attitudes, feelings, and concerns of future teachers towards inclusive education and diversity in the classroom will be analyzed. Furthermore, the relationship of the data with the main analyzed grouping variables will be presented in order to identify possible modulating variables of teacher self-efficacy in the participants. 

### 3.1. Brief Scale of Teacher Self-Efficacy (TSES-SF)

The results of the sample (Table 2) indicate an adequate level of teacher self-efficacy, with differences in minimum and maximum scores within a wide range but within positive values.

#### 3.1.1. Total Teacher Self-Efficacy

On the Total Teacher Self-Efficacy factor, which encompasses the scores of the three aforementioned factors, the results obtained indicate a generally positive sense of teacher self-efficacy. On average, prospective teachers feel that they can “do quite a bit” in effectively using instructional strategies effectively as well as managing classes and engaging students in learning.

Upon analyzing the twelve items (Table 3), however, we find certain weaknesses in the feelings of self-efficacy about the factor of Classroom Management, showing the mean comparison analysis using 7 as the reference value, based on anchor points for a self-efficacy sentiment at levels of “I can do quite a bit” [33], considered the fourth anchor point on the scale, with 1 being “Nothing”, 3 being “Very little”, and 5 being “Something”, with the group means close to this value. These include handling disruptive behavior (M = 6.66, SD = 1.472), calming down a noisy or disruptive student (M = 6.51, SD = 1.585), and creating different classroom management systems for each group (M = 6.78, SD = 1.413). In these aspects, the sample shows greater dispersion and a wider range of responses, this factor being the one that brings together a higher percentage of the sample (34.1%) in the low level of feeling of self-efficacy (Table 4).

The teaching tasks in which the sample demonstrates a higher level of perceived self-efficacy are making students believe they can do the classwork well (item 3-F1) and providing an explanation or alternative example when students are confused (item 10-F2).

Regarding the main grouping variables analyzed, statistically significant differences have been found between groups. Those individuals in the sample who have regular contact with people in situations of special vulnerability exhibited a higher level of total teacher self-efficacy (*t*(303) = 2.765, *p* = 0.006), and particularly those who have experience teaching people in situations of special vulnerability in non-formal contexts (*t*(303) = 3.962, *p* < 0.001). The variable “Motivation to pursue the MDSE” also showed statistically significant differences between groups (*F*(4,300) = 4.858, *p* = 0.001), with higher levels of total teacher self-efficacy observed in individuals who pursued the master’s degree out of vocation compared to their peers who pursued the master’s degree for job security (*p* = 0.001), lack of alternatives (*p* = 0.007), teacher influence (*p* = 0.014), or family influence (Table 5 and Table 6).

Regarding the area of knowledge corresponding to university studies, previous studies show statistically significant differences between groups (*F*(4,300) =2.766, *p*= 0.028), showing a greater feeling of total self-efficacy (Table 7 and Table 8) in people who have conducted studies in the area of Health Sciences versus Engineering and Architecture (*p* = 0.038) and Social and Legal Sciences compared to the latter (*p* = 0.003). These four variables seem to modulate higher levels of teacher self-efficacy in future professionals.

To explore whether variations in the levels and results of the total teacher self-efficacy factor are related to and have a measurable impact on the perception of the training received in the MDSE, attitudes towards diversity and inclusive education, and the feelings and level of concerns of future teachers for their work as educators in an inclusive school, we have employed a few strategies.

We used Pearson’s correlation test to determine if there were significant relationships between total teacher self-efficacy and the training received, assessed through the “Questionnaire for Future Secondary Teachers regarding Perceptions of Diversity (CFDPAD)” (Table 9). The results were statistically significant for dimensions 1, 2, 3, and 4 of the CFDPAD, indicating a positive and continuous relationship in both directions. 

The first dimension of the CFDPAD, conditioning factors in the process of addressing diversity in the classroom (Table 7), refers to attitudinal aspects. This dimension has the strongest correlation with a clear direct and continuous relationship between both variables, and it is therefore significant for future teachers with statistically significant differences between groups (*F*(21,283) = 2.588, *p* < 0.001). These differences highlight that the people with the highest level of total self-efficacy feeling (Table 10 and Table 11) align more with the necessary elements for the development of a quality process of attention to diversity in schools compared to those who are perceived with a low level of teacher self-efficacy (*p* = 0.001 and *p* = 0.010, respectively).

Also, notably, a positive evaluation of the training received in dimension 2 (curricular and organizational response to diversity in the classroom) appears to have the strongest direct and positive relationship with the total teacher self-efficacy and its components. This dimension seems to have the greatest influence on teacher self-efficacy, especially in terms of training for classroom organization, time management, handling of grouping types, use of instructional strategies, diversity-oriented measures and programs, selection and adaptation of objectives, content, and competencies, as well as diversified assessment tasks and activities. This dimension emerges as a key element for future teachers. In the ANOVA analysis, statistically significant differences between groups were found (*F*(23,281) = 2.165, *p* = 0.002), with individuals with a high and medium level of total teacher self-efficacy perceiving greater training in dimension 2 (Table 10 and Table 11) compared to those with a low level of self-efficacy (*p* < 0.001 and *p* = 0.011, respectively).

Regarding dimension 3 of the CFDPAD, teacher training towards diversity, which is directly related to training in the field of special education and diversity in the classroom, similar results were found. In particular, statistically significant differences were observed between groups in the ANOVA analysis (*F*(27,277) = 2.334, *p* < 0.001). Teachers with higher and medium levels of self-efficacy reported a greater perception of their training in handling diversity compared to those with a low level of self-efficacy (Table 10 and Table 11).

Finally, in dimension 4 of the CFDPAD, formative teaching practice in addressing diversity (Table 7), the analysis reveals a positive correlation between the studied variables. This finding is based on evaluations of how well the training received in the MDSE prepares educators to meet the needs of students requiring specific educational support in schools. According to the data (Table 10 and Table 11), with the ANOVA analysis (*F* (14,290) = 2.140, *p* = 0.010), a greater level of training and knowledge to address classroom diversity is related to a higher sense of total teacher self-efficacy, and conversely with significant differences between high and low level (*p* = 0.002).

Lastly, regarding the relationship between this total teacher self-efficacy factor and attitudes, feelings, and concerns towards diversity and inclusive education, the Pearson correlation test (Table 12) reveals significant results concerning all three factors. 

In terms of the factor of concerns about having students with disabilities in the classroom, an inverse relationship is observed. Lower concerns are associated with a higher level of perceived total teacher self-efficacy; in ANOVA analysis, while statistically significant differences appear between groups (*F*(11,293) = 5.218, *p* < 0.001), with participants with low and medium levels of teacher self-efficacy overall presenting perception of a greater number of concerns (Table 13 and Table 14) compared those with a high level of self-efficacy (*p* = 0.001 and *p* = 0.002, respectively). This relationship also applies to the factor of feelings, where lower scores are related to more positive feelings and are associated with a higher score in teacher self-efficacy, with significant differences between groups (*F*(8,296) = 5.098, *p* = 0.001). Lastly, there is a positive direct relationship with factor 1, attitudes, indicating a more positive perception of inclusive education and the concept of students who belong to it with a higher sense of total self-efficacy, and vice versa.

#### 3.1.2. Teacher Self-Efficacy for Student Engagement “Commitment to Students”

This factor refers to the perceived efficacy to engage students in their learning. The overall sample has a positive feeling about their capacity to do so (Table 2). 

Regarding the main analyzed grouping variables, statistically significant differences have been found between groups of women and a higher sense of teacher self-efficacy in “commitment to students” (χ2 (2) = 6.687, *p* = 0.035), with a greater sense of self-efficacy for women in the high level (Table 15), where the score of the corrected standardized residual exceeds 1.96. Other modulating variables include experience in teaching vulnerable individuals in non-formal contexts (*t*(303) = 2.576, *p* = 0.010) and especially the motivation to pursue the MDSE (*F*(4,300) = 5.804, *p* < 0.001), with statistically significant differences between individuals who pursue the master’s out of vocation compared to those who do so for the sake of securing stable employment (*p* = 0.001), those influenced by a family member (*p* = 0.029), and those who enter the master’s program due to lack of better study options (*p* < 0.001), as can be seen in Table 16 and Table 17. Table 18 shows the cross-tabulated data that demonstrate the stark difference between individuals entering the program out of vocation and those seeking stable employment.

Regarding the area of knowledge, there are statistically significant differences between groups (*F*(4,300) =2.605, *p* = 0.036) showing in post hoc analysis, as in the total factor, a greater feeling of self-efficacy for the “Commitment to students” the people who have studied Health Sciences compared to Engineering and Architecture (*p* = 0.033), and Social and Legal Sciences compared to the latter (*p* = 0.008), as can be seen in Table 19 and Table 20.

Respecting the training received and valued through the CFDPAD questionnaire, statistically significant differences have been found between groups in all dimensions and in the same sense as in the factor of total teacher self-efficacy: in all of them, the differences are statistically significant among the sample that presents a high level of self-efficacy “Commitment to students” compared to those who have a low level, the latter feeling worse prepared than the first in all the dimensions valued. Reviewing the results in dimension 1 “Conditioning factors in the process of addressing diversity in the classroom” (*F*(21,283) =2.250, *p* = 0.002) and the differences found in the dimension 5 “Teacher perception towards students with specific educational support needs” (*F*(10,294) = 1.807, *p* = 0.049), not found in the factor total teacher self-efficacy, it is observed that the future teachers with a high level of self-efficacy “commitment to students” exhibit a more positive attitude both towards the attention to diversity as to the response to educational needs in the classroom to general level and a higher level of agreement on the elements that should be present in the teaching and learning process for the development of quality diversity in the classrooms compared to those with a low (*p* < 0.001) or medium (*p* = 0.008) level, as can be seen in Table 21 and Table 22.

Regarding the impact of this self-efficacy factor “commitment to students” on attitudes, feelings, and concerns towards diversity and inclusive education, in the ANOVA analysis, we found significant differences between groups in the feelings (*F*(8,296) = 3.891, *p* = 0.005) and concerns (*F*(11,293) = 4.865, *p* = 0.016) factors, showing in both factors fewer worries and fewer negative feelings at a higher level of self-efficacy for the “commitment to students” and vice versa. The post hoc analysis shows worse feelings (higher scores) in people with a low level of self-efficacy compared to medium (*p* = 0.033) and high (*p* = 0.001) levels, and greater concerns about having diverse students in their classrooms in the people with low and medium levels of self-efficacy for “commitment to students” compared to those with a high level with *p* < 0.001 and *p* = 0.006, respectively (Table 23 and Table 24).

#### 3.1.3. Teacher Self-Efficacy to Optimize One’s Own Instruction “Instructional Strategies”

This factor refers to the perceived efficacy to utilize different instructional strategies that are suitable for classroom diversity. The overall sample has a positive perception of this factor (Table 2), with it being the highest-rated factor in the TSES-SF scale. 

Regarding the main analyzed grouping variables, statistically significant differences have been found between groups, with a higher sense of self-efficacy in “instructional strategies” observed in individuals with experience in teaching vulnerable individuals in non-formal contexts (*t* (303) = 2.712, *p* = 0.007). The variable of motivation to pursue the MDSE re-emerges, showing significant statistical differences (*F*(4,300) =3.021, *p* = 0.018) between individuals who pursue the master’s out of vocation compared to those who do so for the sake of securing stable employment (*p* = 0.006) and those who take it because they do not have a better option (*p* = 0.044), as can be seen in Table 25 and Table 26. Finally, there are statistically significant differences between the students who have completed the MDSE in a public university versus a private university (*t* (303) = −2.437, *p* = 0.015) feeling the second-highest level of self-efficacy for the development of “instructional strategies”.

Regarding the relationship with the training received in the MDSE, it is found that this factor influences both dimension 1 “Conditioning elements of the process of attention to diversity in the classroom” (*F*(21,283) = 2.596, *p* = 0.023), which is more related to attitudes towards inclusive education, as well as dimensions 2 “Curricular and organizational response to diversity in the classroom” (*F*(23,281) = 1.724, *p* = 0.023), 3 “Teacher training towards diversity” (*F*(27,277) = 2.069, *p* = 0.002), and 5 “Teacher perception towards students with specific educational support needs” (*F*(10,294) = 2.571, *p* = 0.005), which are directly related to the training received in the MDSE (Table 27).

Regarding dimension 1, statistically significant differences are observed (Table 28) between the groups with a high and medium level of self-efficacy and the group with a low level (*p* = 0.002 and *p* = 0.042, respectively). For dimension 2, significant differences are found (Table 28) between individuals with a high level of self-efficacy and those with both low a level (*p* = 0.015) and a medium level (*p* = 0.005). 

Finally, significant differences between groups are found in all dimensions of the SACIE-R scale (Table 29): “Feelings” (*F*(8,296) = 4.754, *p* < 0.001) and “Concerns” (*F*(11,293) = 3.175, *p* < 0.001). As shown in Table 30, individuals with a low level of self-efficacy in instructional strategies exhibit higher levels of negative feelings compared to those with a high level of self-efficacy (*p* = 0.030), as well as those who have a medium level of self-efficacy exhibiting more concerns in instructional strategies compared to those with a high level of self-efficacy in this factor (*p* = 0.002).

#### 3.1.4. Teacher Self-Efficacy in Classroom Management “Classroom Management”

In this last factor, the scores are the lowest in the entire scale, making it the weak point of the analyzed sample (Table 2). Managing disruptive behavior (item 1), the situation of having to calm down a noisy or disruptive student (item 7), and establishing a classroom management system with each group (item 8) are the situations for which the sample overall shows the least efficacy (Table 3), also showing a wider range of responses.

Regarding the main grouping variables analyzed, statistically significant differences have been found between groups, showing a higher sense of self-efficacy in “classroom management” between groups from different areas of knowledge (*F*(4,300) = 3.203, *p* = 0.013), with individuals in the Social and Legal Sciences feeling more capable (Table 31 and Table 32) compared to those in Engineering- and Architecture-related fields (*p* = 0.002) and Arts and Humanities (*p* = 0.032). Finally, statistically significant differences appear again between groups in the variable habitual contact with people in a situation of special vulnerability (*t* (303) = 3.476, *p* = 0.001), showing a greater feeling of self-efficacy the people who maintain this contact. This also happens with the variable experience in non-formal education for people in situations of special vulnerability (*t* (303) = 4.942, *p* > 0.001).

The perceived level of self-efficacy in classroom management in this factor appears to have a significant impact within the CFDPAD on dimension 1 “Determining Elements of Diversity in the Classroom” (*F*(21,283) = 2.724, *p* < 0.001). Individuals with higher perceived self-efficacy in this area show greater agreement on the premises of inclusive education and the need for key elements to be implemented in the educational setting. Significant differences also appear in the ANOVA analysis in dimensions 2 “Curricular and organizational response to diversity in the classroom” (*F*(23,281) = 2.767, *p <* 0.001), 3 (*F*(27,277) = 1.963, *p =* 0.004), and 4 “Practice training teacher in attention to diversity” (*F*(14,290) = 2.090, *p* = 0.012), with significant differences in post hoc analysis between high versus medium and low levels in both (Table 33 and Table 34).

Lastly, in relation to the SACIE-R scale, the level of self-efficacy in classroom management shows significant differences between groups within the “concerns” dimension (*F* (11,293) = 4.896, *p* < 0.001), “feelings” (*F* (8,296) = 4.573; *p* < 0.001), and “attitudes” (*F* (14,290) = 2.139; *p* = 0.010). Individuals with a low and medium level of self-efficacy express greater concerns compared to those who feel more capable of managing the classroom (*p* < 0.001). Furthermore, those with a low level have a greater score in the dimension of feelings compared to those who present a high feeling of self-efficacy (*p* = 0.013), showing worse feelings towards inclusive education and the students involved in it (Table 35 and Table 36).

## 4. Discussion

Firstly, it should be noted that, in a systematic review on studies from 2015 to 2021 [1] related to the sense of self-efficacy in future teachers of the master’s degree, no previous work was identified on the relationship between the self-efficacy and the training received in the master’s degree. As such, the current research contributes to a gap in the literature related to teacher training for secondary education in inclusive schools, its influence on the relevant perception of self-efficacy to face this challenge, and its ultimate connection with workplace well-being, including teacher well-being as a pillar of sustainable well-being in today’s schools.

On the other hand, there are numerous studies focused on teacher self-efficacy in relation to inclusion, diversity, and classroom management. These research endeavors underscore the significance of teacher training and ongoing professional development in enhancing self-efficacy and fostering favorable attitudes towards inclusion and diversity [48,49]. For instance, one study discovered that educators who underwent professional development in inclusive education and special educational needs exhibited more favorable attitudes towards inclusion [48]. Another study indicated that both initial and continuous training in classroom management positively influenced teacher self-efficacy in this area [49].

The main result of this study indicates that the level of teacher self-efficacy among future secondary education teachers in our sample is moderate to high. In general, they have confidence in their ability to effectively use instructional strategies, manage the classroom, and engage students in learning. This is highly positive, as various authors have noted [1,50], since higher levels of self-efficacy are associated with increased motivation and performance in the teaching profession. It is also related to improved student outcomes [51], a better work environment, innovation, and sustainable well-being [19,20,52]. Furthermore, teachers with high levels of self-efficacy set goals for themselves and exert greater effort to achieve them, which has direct implications for their teaching practice and forms the basis for their subsequent judgments of self-efficacy in a cyclical process [53,54].

The sample particularly demonstrates a strong sense of efficacy in the dimension of teacher self-efficacy to optimize their own instruction through instructional strategies. This finding is relevant considering the research conducted by other authors on the relationship between teacher self-efficacy and the development of inclusive strategies in the classroom [54]. These heightened feelings of self-efficacy, positively linked to instructional effectiveness, prompt teachers to engage in organizational planning and demonstrate a proactive approach towards the challenges and obstacles encountered in their daily teaching. Moreover, teacher self-efficacy can mitigate and alleviate acute stressors associated with work, thereby enhancing teachers’ overall well-being [55].

A positive classroom discipline climate [56] is the aspect that most significantly affects the perception of secondary school teachers regarding their self-efficacy. These results align with the conclusions of other authors [57,58] and highlight the need for enhanced initial training on teaching and learning processes and classroom management. The training should promote the adoption of norms, conflict resolution, and self-management and self-control skills, thereby increasing the future teachers’ ability to create positive learning environments and their sense of self-efficacy [59,60,61].

This weakness, which appears to be more pronounced in the sample of future teachers in the field of Engineering and Architecture compared to those in the field of Social and Legal Sciences or Health Sciences, aligns with the findings of other similar studies [62]. In the long run, it may have a dual effect. On one hand, it can discourage teachers, negatively impacting their psychological well-being, and on the other hand, it may devalue the teaching of subject knowledge in favor of focusing solely on discipline management, thereby disregarding the objectives of comprehensive student development and attention to diversity.

The significance of these findings cannot be overstated, as the examination of teacher self-efficacy and its implications for sustainable workplace well-being highlight its pivotal role in shaping educators’ perceptions of their profession, classroom dynamics, and their influence on student learning outcomes. Consequently, educators with heightened self-efficacy are inclined to adopt effective pedagogical strategies and techniques, leading to heightened student engagement and academic success. Furthermore, teacher self-efficacy directly impacts performance, reflecting overall competence [63].

But what variables have modulated the feeling of teaching self-efficacy, both at a global level and in each of its factors? The perception of teacher self-efficacy is influenced by two types of factors [64]: direct factors related to the actions teachers take in their daily interactions with students, and indirect factors such as vocation, personality traits, mastery of specific content or skills, and attitudes towards students and inclusive schooling, among others. In our sample, vocation and other indirect factors have been key contributors to an elevated perception of teacher self-efficacy.

These findings are also supported by other authors [56], who note that pre-service teachers with a strong vocational inclination tend to have higher levels of teacher self-efficacy. Furthermore, the review of Bueno-Alvarez [1] highlights how intrinsic goals in MUPES, which could be linked to vocation as the reason for pursuing studies, exert a positive, direct, and indirect effect through academic autonomous motivation on teacher self-efficacy. Conversely, extrinsic goal contents revealed a direct and negative effect on teacher self-efficacy in MUPES students [10,65,66]. In addition, it is important to understand the significance of vocational inclination, along with training, in shaping teachers’ perception of the context in which they operate, whether as a challenge or a threat. Likewise, the vocation constitutes an essential condition for achieving job satisfaction in the teaching profession [67,68], which ultimately leads to high levels of motivation and low work stress [69].

The variable of regular and close contact with individuals in situations of special vulnerability, which is one of the most influential factors in attitudes and perceptions of inclusion in today’s schools [68], may also be related to the sense of teacher self-efficacy in inclusive schools due to the positive attitude it fosters towards inclusive education. Regarding the factors analyzed, this *contact* appears as a particularly significant variable in the development of feeling of self-efficacy for classroom management and class management.

The most influential variable found in relation to the level of teacher self-efficacy, namely experience in non-formal teaching to individuals in situations of special vulnerability [70], has allowed future teachers to develop ways of “doing and being” [71] with diversity in educational processes prior to their MDSE program. Practical training received by future teachers influences their sense of professional self-efficacy through vicarious experiences, and as previous research indicates, other variables such as prior experiences also play a role [1].

Furthermore, training in the working context, as argued by Levy-Leboyer [72], surpasses any form of education, as “experiences gained from action, assuming real responsibility, and facing concrete problems truly provide competencies that even the best instruction cannot deliver”. On the other hand, recent studies [73] show how one can observe a more positive influence in teachers associated with these previous experiences working with boys and girls with special education needs informally, rather than having learned through regulated training.

Within the specific factor related to the commitment to students, women exhibit a higher sense of self-efficacy compared to men, which is consistent with other studies that have shown similar results associated with the underlying interpersonal intelligence [74]. Additionally, there are studies investigating variations in teacher self-efficacy based on gender and university type. For instance, one study discovered that female teachers exhibited greater self-efficacy in classroom management compared to their male counterparts [75]. Another study revealed that educators from private universities demonstrated higher self-efficacy in instructing students with disabilities than those from public universities [49].

In the present investigation, there appear to be statistically significant differences between the MDSE students at private universities, who feels more capable of optimizing their own instruction through the development of various educational strategies, and the MSDE students at public universities. These data show similarities with those found by Sebastiá et al. [76] in a study on the monitoring and accreditation reporting processes of MUPES in Spain. In the comparative analysis between private and public universities, the former had more positive evaluations, achieving higher scores than the public ones in most of the evaluated criteria. It would be necessary to conduct a broader analysis in new research, and on a more representative sample, to corroborate these data and to understand, if applicable, the indicators or criteria that may be behind these results for the continuous improvement of institutions.

As previously mentioned, for effective teaching action [21,22] it is not enough to have a sense of self-worth. It is also essential to have knowledge of the subject matter and mastery of a set of competencies and skills, including those indicated in the Profile of the Inclusive Teacher [17]. Among these, the named dimensions are key, as they indicate the preparation of future teachers for Universal Design for Learning [77] and attention to diversity within the framework of inclusive education. We cannot ignore that, in terms of the correlation between self-efficacy and pursuing a master’s program, there is evidence suggesting that such programs can have a positive influence on teacher self-efficacy. For example, a study revealed that educators who engaged in a master’s program in special education experienced a boost in their self-efficacy when teaching students with disabilities [49].

No statistically significant differences have been found among age groups, with the initially striking observation being the large percentage of students over the age of 31. This result is also found in a recent study [78] with students from the MDSE, where the average age was 34.49 years, with 71.2% of individuals being 30 years or older. Furthermore, upon examining data from the Ministry of Universities [79], it is found that 38.56% of students enrolled in the MDSE program this academic year are 31 years or older.

On the other hand, we cannot separate these findings from the period of practical training, knowing that the practical training of future teachers significantly influences their sense of self-efficacy [1], depending on their level of competence, intrinsic motivation, self-concept, and regulatory capacity [80]. This is relevant because the practical training, particularly the period of practical experience and the guidance provided by mentors in the schools, should ensure the observation and modeling of behaviors and attitudes of self-efficacy in future teachers. It is important to be aware that we cannot expect desired behaviors if we do not model and provide feedback, emphasizing the importance of vicarious learning and feedback [71].

Additionally, an inverse relationship was found between the concerns factor and the level of self-efficacy, suggesting that lower perceived self-efficacy is associated with greater concerns about addressing diversity. As indicated by several authors [81], one of the areas with the greatest impact on the development of a sustainable school system is “from the well-being of individuals to the well-being of the community”. This concept of well-being extends to teachers, given their critical role in influencing the well-being of other stakeholders. The connection between teacher well-being, their training for inclusive schooling, its underlying values, and their sense of self-efficacy is fundamental.

These findings are consistent with other studies that demonstrate strong relationships between feelings, attitudes, concerns, and perceptions of teacher self-efficacy in the development of inclusive education [82,83]. In addition, the lower level of concerns, associated with higher well-being, also aligns with the analysis conducted by other authors on teacher self-efficacy and its effect on sustainable well-being. This demonstrates that teacher self-efficacy significantly shapes educators’ perceptions of their roles, classroom dynamics, and their influence on student learning outcomes [63].

In summary, there exists a substantial body of research centered on teacher self-efficacy concerning inclusion, diversity, well-being, and classroom management. These studies underscore the significance of teacher training and professional development in enhancing self-efficacy and fostering positive attitudes towards inclusion and diversity in sustainable schools [14,84].

## 5. Conclusions

This study provides relevant results concerning the sense of self-efficacy in future teachers of the master’s program (MDSE) in relation to the training received in the MDSE. This was made possible by the use of the Questionnaire for Future Secondary Education Teachers Regarding Perceptions of Diversity Attention, developed in Spain specifically to evaluate the impact of the training received in the master’s program on the students.

The findings show that, overall, future teachers have an adequate level of teacher self-efficacy, although they exhibit specific weaknesses in certain aspects. Generally, participants feel capable of using effective instructional strategies and managing classes but show less confidence in handling disruptive behavior and calming noisy or disruptive students.

Significant differences in teacher self-efficacy were observed according to various variables, such as experience in teaching people in situations of special vulnerability, motivation to pursue the master’s degree, and participants’ area of knowledge.

Additionally, a positive correlation was found between teacher self-efficacy and the perception of the training received in the master’s program, as well as with attitudes towards diversity and inclusive education. In general, those with higher levels of teacher self-efficacy also showed a more positive attitude towards diversity in the classroom and less concern about the special needs of students. And this correlation found is highly valuable, emphasizing its importance not only for the improvement of educational inclusion processes and sustainable development but especially for pinpointing relevant areas of improvement in MDSE studies.

With respect to the three specific factors of teacher self-efficacy (commitment to students, instructional strategies, and classroom management), similar patterns of results and correlations with demographic and training variables were observed. In summary, the study highlights the importance of teacher self-efficacy in the preparation of future educators for inclusive and sustainable schools, and its relationship with the training received, attitudes towards diversity and inclusive education, and effective classroom management.

Regarding some of the practical implications of the results, the following suggestions are made with the aim of supporting the professional and personal development [17] of future secondary school teachers: (1) To improve teacher training in discipline management and the strengthening of moral authority. This will enable teachers to inspire enthusiasm, resolve conflicts fairly, transmit a passion for learning, and ultimately foster the holistic development of students [85]. (2) To promote action learning and reflection in practice centers to enhance teaching practice. The selection of practice centers and mentors should be considered [86,87], taking into account the criteria recommended by research in the field of inclusion and educational innovation and being aware of the importance of this selection. The observation of other colleagues and the verbal persuasion that can be received from these are two key elements for improving teacher self-efficacy. (3) To continue researching teacher self-efficacy and MDSE training to identify the elements acquired during the received training. This will allow intentional programming within training programs, along with practical activities, to promote the self-efficacy of trainee teachers as presented in other studies; this is important, given that beliefs about teaching effectiveness are malleable in the first stages of learning. (4) To foster the engagement of future teachers with the most vulnerable students in the educational system, in a diverse environment, through collaboration with educational institutions, entities, and associations. This can be achieved through the organization of volunteers in universities, from the beginning of the previous university degrees, both in scientific as well as social and artistic fields, and through the development of Service-Learning projects and other active and social methodologies in MDSE subjects. (5) To promote understanding within the MDSE about the importance of sustainable well-being as one of the values to support the construction of inclusive education [14]. (6) There should be a mentoring program to ensure that the beginning teacher develops contextual skills, reflection skills, and as such learns from their practice—especially in the first few years of their teaching.

## Figures and Tables

**Table 1 behavsci-14-00563-t001:** General characteristics of the sample.

Study Variables		*n*	%
Motivation to study the MDSE.	Vocation	133	43.6
	Possibility of finding a stable job	126	41.3
	For lack of a better option in my career path	14	4.6
	Influenced by a teacher who has influenced me in my education	16	5.2
	Due to the influence of a family member who is or has been a teacher	16	5.2
Age	25 years old or less	73	23.9
	Between 26 and 31 years old	86	28.2
	Older than 31 years old	146	47.9
Area of knowledge of higher studies of access to MDSE.	Health Sciences	8	2.6
	Science	53	17.4
	Engineering and Architecture	59	19.3
	Social and Legal Sciences	85	27.9
	Arts and Humanities	100	32.8

**Table 2 behavsci-14-00563-t002:** Descriptive statistics teacher self-efficacy TSES-SF.

	*n*	Mean	SD	Min.	Max.
Total Self-Efficacy	305	7.10	0.972	4	9
Self-Efficacy F1 Commitment to students	305	7.25	0.993	4	9
Self-Efficacy F2 Instructional Strategies	305	7.31	1.065	4	9
Self-Efficacy F3 Classroom Management	305	6.75	1.236	2	9

**Table 3 behavsci-14-00563-t003:** Descriptive statistics and t-distribution of the sample.

TSES-SF Items	Mean	SD	Min.	Max.	t	*p*
**Self-Efficacy F1 Commitment to Students**						
**Item 2.** F1 What do you believe you will be able to do to motivate students who show little interest in class?	7.09	1.264	2	9	1.178	0.240
**Item 3.** F1 What do you believe you will be able to do to make students believe they can perform well in class?	7.64	1.046	3	9	10.679	0.000
**Item 4.** F1 What do you believe you will be able to do to help your students value learning?	7.22	1.296	3	9	2.961	0.003
**Item 11.** F1 What do you believe you will be able to do to advise families to help their children perform well in school?	7.04	1.459	1	9	0.471	0.638
**Self-Efficacy F2 Instructional Strategies**						
**Item 5.** F2 What do you believe you will be able to do to teach students to ask good questions?	7.18	1.250	3	9	2.519	0.012
**Item 9.** F2 What do you believe you will be able to do to use a variety of assessment strategies?	7.06	1.514	2	9	0.681	0.497
**Item 10.** F2 What do you believe you will be able to do to provide an alternative explanation or example when students are confused?	7.80	1.318	1	9	10.643	0.000
**Item 12.** F2 What do you believe you will be able to do to implement alternative strategies in your class?	7.22	1.318	1	9	2.868	0.004
**Self-Efficacy F3 Classroom Management**						
**Item 1.** F3 What do you believe you will be able to do to manage disruptive behavior in class?	6.66	1.472	1	9	−4.084	0.000
**Item 6.** F3 What do you believe you will be able to do to get students to follow class rules?	7.05	1.333	1	9	0.644	0.520
**Item 7.** F3 What do you believe you will be able to do to calm down a disruptive or noisy student?	6.51	1.585	1	9	−5.345	0.000
**Item 8.** F3 What do you believe you will be able to do to establish a classroom management system with each group?	6.78	1.413	1	9	−2.756	0.006

**Table 4 behavsci-14-00563-t004:** Frequencies of perceived levels of teachers’ self-efficacy.

TSES-SF Levels		*n*	%
Total Self-Efficacy	Low	66	21.6
	Medium	127	41.6
	High	112	36.7
	Total	305	100.0
Self-Efficacy F1 Commitment to Students	Low	52	17.0
	Medium	111	36.4
	High	142	46.6
	Total	305	100.0
Self-Efficacy F2 Instructional Strategies	Low	51	16.7
	Medium	97	31.8
	High	157	51.5
	Total	305	100.0
Self-Efficacy F3 Classroom Management	Low	104	34.1
	Medium	115	37.7
	High	86	28.2
	Total	305	100.0

**Table 5 behavsci-14-00563-t005:** ANOVA analysis of motivation to study the MDSE.

ANOVA					
Total Teacher Self-Efficacy					
	Sum of Squares	df	RMSE	F	Sig.
Between groups	17.460	4	4.365	4.858	0.001
Within groups	269.550	300	0.899		
Total	287.011	304			

**Table 6 behavsci-14-00563-t006:** Post hoc DMS analysis of motivation to study the MDSE.

Multiple Comparisons						
Dependent Variable: Total Teacher Self-Efficacy						
(I) Motivation	(J) Motivation	Mean Difference (I–J)	Std. Error	Sig.	95% Confidence Interval	
					Lower Bound	Upper Bound
**Vocation**	Possibility of finding a stable job	0.381 *	0.118	0.001	0.15	0.61
	For lack of a better option in my career path	0.720 *	0.266	0.007	0.20	1.24
	Influenced by a teacher who has influenced me in my education	0.617 *	0.251	0.014	0.12	1.11
	Due to the influence of a family member who is or has been a teacher	0.580 *	0.251	0.021	0.09	1.07

* The mean difference is significant at the 0.05 level.

**Table 7 behavsci-14-00563-t007:** ANOVA analysis of area of knowledge of higher studies of access to MDSE.

ANOVA					
Total Teacher Self-Efficacy					
	Sum of Squares	df	RMSE	F	Sig.
Between groups	10.207	4	2.552	2.766	0.028
Within groups	276.803	300	0.923		
Total	287.011	304			

**Table 8 behavsci-14-00563-t008:** Post hoc DMS analysis of area of knowledge of higher studies of access to MDSE.

Multiple Comparisons						
Dependent Variable: Total Teacher Self-Efficacy						
(I) Area of Knowledge	(J) Area of Knowledge	Mean Difference (I–J)	Std. Error	Sig.	95% Confidence Interval	
					Lower Bound	Upper Bound
**Health Sciences**	Science	0.509	0.364	0.163	−0.21	1.23
	Engineering and Architecture	0.754 *	0.362	0.038	0.04	1.47
	Social and Legal Sciences	0.271	0.355	0.447	−0.43	0.97
	Arts and Humanities	0.519	0.353	0.142	−0.18	1.21
**Social and Legal Sciences**	Science	0.239	0.168	0.156	−0.09	0.57
	Health Sciences	−0.271	0.355	0.447	−0.97	0.43
	Engineering and Architecture	0.484 *	0.163	0.003	0.16	0.80
	Arts and Humanities	0.249	0.142	0.080	−0.03	0.53

* The mean difference is significant at the 0.05 level.

**Table 9 behavsci-14-00563-t009:** Correlations between training received in the MDSE and total teacher self-efficacy.

CFDPAD Dimensions					
		D1 “Conditioning Factors in the Process of Addressing Diversity in the Classroom”	D2 “Curricular and Organizational Response to Diversity in the Classroom”	D3 “Teacher Training towards Diversity”	D4 “Formative Teaching Practice in Addressing Diversity”
**Total Self-Efficacy**	C. Correlation	0.266 **	0.265 **	0.198 **	0.216 **
	*p*	<0.001	<0.001	0.001	<0.001

** The correlation is significant at the 0.001 level (two-tailed).

**Table 10 behavsci-14-00563-t010:** ANOVA analysis of CFDPAD dimensions.

ANOVA						
		Sum of Squares	df	RMSE	F	Sig.
**D1: “** **Conditioning factors in the process of addressing diversity in the classroom”**	Between groups	46.236	21	2.202	2.588	0.000
	Within groups	240.775	283	0.851		
	Total	287.011	304			
**D2: “** **Curricular and organizational response to diversity in the classroom”**	Between groups	43.201	23	1.878	2.165	0.002
	Within groups	243.810	281	0.868		
	Total	287.011	304			
**D3: “** **Teacher training towards diversity”**	Between groups	53.185	27	1.970	2.334	0.000
	Within groups	233.826	277	0.844		
	Total	287.011	304			
**D4: “** **Formative teaching practice in addressing diversity”**	Between groups	26.877	14	1.920	2.140	0.010
	Within groups	260.134	290	0.897		
	Total	287.011	304			

**Table 11 behavsci-14-00563-t011:** Post hoc DMS analysis of CFDPAD dimensions.

Multiple Comparisons							
Dependent Variable	(I) Level of Teachers’ Self-Efficacy	(J) Level of Teachers’ Self-Efficacy	Mean difference (I–J)	Std. Error	Sig.	95% Confidence Interval	
						Lower Bound	Upper Bound
**D1: “** **Conditioning factors in the process of addressing diversity in the classroom”**	Low	Medium	−0.05080	0.03978	0.203	−0.1291	0.0275
		High	−0.13904 *	0.04068	0.001	−0.2191	−0.0590
	Medium	Low	0.05080	0.03978	0.203	−0.0275	0.1291
		High	−0.08824 *	0.03398	0.010	−0.1551	−0.0214
	High	Low	0.13904 *	0.04068	0.001	0.0590	0.2191
		Medium	0.08824 *	0.03398	0.010	0.0214	0.1551
**D2: “** **Curricular and organizational response to diversity in the classroom”**	Low	Medium	−0.29595 *	0.11553	0.011	−0.5233	−0.0686
		High	−0.43273 *	0.11815	0.000	−0.6652	−0.2002
	Medium	Low	0.29595 *	0.11553	0.011	0.0686	0.5233
		High	−0.13678	0.09869	0.167	−0.3310	0.0574
	High	Low	0.43273 *	0.11815	0.000	0.2002	0.6652
		Medium	0.13678	0.09869	0.167	−0.0574	0.3310
**D3: “** **Teacher training towards diversity”**	Low	Medium	−0.24974 *	0.10931	0.023	−0.4648	−0.0346
		High	−0.27363 *	0.11178	0.015	−0.4936	−0.0537
	Medium	Low	0.24974 *	0.10931	0.023	0.0346	0.4648
		High	−0.02389	0.09338	0.798	−0.2076	0.1599
	High	Low	0.27363 *	0.11178	0.015	0.0537	0.4936
		Medium	0.02389	0.09338	0.798	−0.1599	0.2076
**D4: “** **Formative teaching practice in addressing diversity”**	Low	Medium	−0.14913	0.07758	0.056	−0.3018	0.0035
		High	−0.24248 *	0.07934	0.002	−0.3986	−0.0864
	Medium	Low	0.14913	0.07758	0.056	−0.0035	0.3018
		High	−0.09335	0.06627	0.160	−0.2238	0.0371
	High	Low	0.24248 *	0.07934	0.002	0.0864	0.3986
		Medium	0.09335	0.06627	0.160	−0.0371	0.2238

* The mean difference is significant at the 0.05 level.

**Table 12 behavsci-14-00563-t012:** Correlation between SACIE-R factors and total teacher self-efficacy.

		F1 Attitudes	F2 Feelings	F3 Concerns
**Total Self-Efficacy TSES-SF**	C. Correlation	0.114 *	−0.161 **	−0.272 **
	*p*	0.047	0.005	<0.001

* The correlation is significant at the 0.05 level (two-tailed). ** The correlation is significant at the 0.001 level (two-tailed).

**Table 13 behavsci-14-00563-t013:** ANOVA analysis of SACIE-R factors.

ANOVA						
		Sum of Squares	df	RMSE	F	Sig.
**F1 Attitudes**	Between groups	24.445	14	1.746	1.928	0.023
	Within groups	262.566	290	0.905		
	Total	287.011	304			
**F2 Feelings**	Between groups	34.759	8	4.345	5.098	0.000
	Within groups	252.251	296	0.852		
	Total	287.011	304			
**F3 Concerns**	Between groups	47.017	11	4.274	5.218	0.000
	Within groups	239.993	293	0.819		
	Total	287.011	304			

**Table 14 behavsci-14-00563-t014:** Post hoc DMS analysis of F3 SACIE-R concerns.

Multiple Comparisons							
Dependent Variable	(I) Level of Teachers’ Self-Efficacy	(J) Level of Teachers’ Self-Efficacy	Mean Difference (I–J)	Std. Error	Sig.	95% Confidence Interval	
						Lower Bound	Upper Bound
**F3 Concerns**	Low	Medium	0.07197	0.10009	0.473	−0.1250	0.2689
		High	0.33313 *	0.10236	0.001	0.1317	0.5346
	Medium	Low	−0.07197	0.10009	0.473	−0.2689	0.1250
		High	0.26116 *	0.08550	0.002	0.0929	0.4294
	High	Low	−0.33313 *	0.10236	0.001	−0.5346	−0.1317
		Medium	−0.26116 *	0.08550	0.002	−0.4294	−0.0929

* The mean difference is significant at the 0.05 level.

**Table 15 behavsci-14-00563-t015:** Cross-tabulation between the variable gender and the level of self-efficacy in “Commitment to students”.

			Self-Efficacy Commitment to Students			Total
			Low	Medium	High	
**Sex**	**Female**	Count				205
		Expected count	35.0	74.6	95.4	205.0
		% within gender	15.6%	32.7%	51.7%	100.0%
		Corrected residual	−1.0	−1.9	2.6	
	**Male**	Count	20			100
		Expected count	17.0	36.4	46.6	100.0
		% within gender	20.0%	44.0%	36.0%	100.0%
		Corrected residual	1.0	1.9	−2.6	
Total		Count				305
		Expected count	52.0	111.0	142.0	305.0
		% within gender	17.0%	36.4%	46.6%	100.0%

**Table 16 behavsci-14-00563-t016:** ANOVA analysis of motivation to study the MDSE.

ANOVA					
Teacher Self-Efficacy Commitment to Students					
	Sum of Squares	df	RMSE	F	Sig.
Between groups	21.543	4	5.386	5.804	0.000
Within groups	278.390	300	0.928		
Total	299.932	304			

**Table 17 behavsci-14-00563-t017:** Post hoc DMS analysis of motivation to study the MDSE.

Multiple Comparisons						
Dependent Variable: Teacher Self-Efficacy Commitment to Students						
(I) Motivation	(J) Motivation	Mean Difference (I–J)	Std. Error	Sig.	95% Confidence Interval	
					Lower Bound	Upper Bound
**Vocation**	Possibility of finding a stable job	0.398 *	0.120	0.001	0.16	0.63
	For lack of a better option in my career path	0.995 *	0.271	0.000	0.46	1.53
	Influenced by a teacher who has influenced me in my education	0.529 *	0.255	0.039	0.03	1.03
	Due to the influence of a family member who is or has been a teacher	0.560 *	0.255	0.029	0.06	1.06

* The mean difference is significant at the 0.05 level.

**Table 18 behavsci-14-00563-t018:** Motivation to study the MDSE and level of self-efficacy in “Commitment to students”.

Self-Efficacy Commitment to Students					
		Low	Medium	High	Total
**Motivation**	Vocation	16	34	83	133
	Possibility of securing stable employment	23	54	49	126
	Lack of better options in my career path	6	8	0	14
	Influence of a teacher who has had a significant impact on my education	1	11	4	16
	Influence of a family member who is or has been a teacher	6	4	6	16
Total		52	111	142	305

**Table 19 behavsci-14-00563-t019:** ANOVA analysis of area of knowledge of higher studies of access to MDSE.

ANOVA					
Teacher Self-Efficacy Commitment to Students					
	Sum of Squares	df	RMSE	F	Sig.
Between groups	10.070	4	2.517	2.605	0.036
Within groups	289.863	300	0.966		
Total	299.932	304			

**Table 20 behavsci-14-00563-t020:** Post hoc DMS analysis of area of knowledge of higher studies of access to MDSE.

Multiple Comparisons						
Dependent Variable: Teacher Self-Efficacy Commitment to Students						
(I) Area of Knowledge	(J) Area of Knowledge	Mean Difference (I–J)	Std. Error	Sig.	95% Confidence Interval	
					Lower Bound	Upper Bound
**Health sciences**	Science	0.638	0.373	0.088	−0.10	1.37
	Engineering and Architecture	0.796 *	0.370	0.033	0.07	1.52
	Social and Legal Sciences	0.354	0.364	0.331	−0.36	1.07
	Arts and Humanities	0.620	0.361	0.087	−0.09	1.33
**Social and Legal Sciences**	Science	0.284	0.172	0.099	−0.05	0.62
	Health Sciences	−0.354	0.364	0.331	−1.07	0.36
	Engineering and Architecture	0.442 *	0.167	0.008	0.11	0.77
	Arts and Humanities	0.266	0.145	0.067	−0.02	0.55

* The mean difference is significant at the 0.05 level.

**Table 21 behavsci-14-00563-t021:** ANOVA analysis of CFDPAD dimensions.

ANOVA						
		Sum of Squares	df	RMSE	F	Sig.
**D1: “** **Conditioning factors in the process of addressing diversity in the classroom”**	Between groups	42.910	21	2.043	2.250	0.002
	Within groups	257.022	283	0.908		
	Total	299.932	304			
**D5: “** **Teacher perception towards students with specific educational support needs”**	Between groups	17.371	10	1.737	1.807	0.049
	Within groups	282.562	294	0.961		
	Total	299.932	304			

**Table 22 behavsci-14-00563-t022:** Post hoc DMS analysis of dimension 1 of the CFDPAD.

Multiple Comparisons							
Dependent Variable	(I) Level of Teachers’ Self-Efficacy in Commitment to Students	(J) Level of Teachers’ Self-Efficacy in Commitment to Students	Mean Difference (I–J)	Std. Error	Sig.	95% Confidence Interval	
						Lower Bound	Upper Bound
**D1: “** **Conditioning factors in the process of addressing diversity in the classroom”**	Low	Medium	−0.06060	0.04393	0.169	−0.1471	0.0259
		High	−0.14926 *	0.04238	0.000	−0.2326	−0.0659
	Medium	Low	0.06060	0.04393	0.169	−0.0259	0.1471
		High	−0.08866 *	0.03312	0.008	−0.1538	−0.0235
	High	Low	0.14926 *	0.04238	0.000	0.0659	0.2326
		Medium	0.08866 *	0.03312	0.008	0.0235	0.1538

* The mean difference is significant at the 0.05 level.

**Table 23 behavsci-14-00563-t023:** ANOVA analysis of SACIE-R factors.

ANOVA						
		Sum of Squares	df	RMSE	F	Sig.
**F2 Feelings**	Between groups	28.543	8	3.568	3.891	0.000
	Within groups	271.389	296	0.917		
	Total	299.932	304			
**F3 Concerns**	Between groups	46.317	11	4.211	4.865	0.000
	Within groups	253.615	293	0.866		
	Total	299.932	304			

**Table 24 behavsci-14-00563-t024:** Post hoc DMS analysis of SACIE-R factors.

Multiple Comparisons							
Dependent Variable	(I) Level of Teachers’ Self-Efficacy in Commitment to Students	(J) Level of Teachers’ Self-Efficacy in Commitment to Students	Mean Difference (I–J)	Std. Error	Sig.	95% Confidence Interval	
						Lower Bound	Upper Bound
**F2 Feelings**	Low	Medium	0.19577 *	0.09137	0.033	0.0160	0.3756
		High	0.29252 *	0.08813	0.001	0.1191	0.4659
	Medium	Low	−0.19577 *	0.09137	0.033	−0.3756	−0.0160
		High	0.09675	0.06888	0.161	−0.0388	0.2323
	High	Low	−0.29252 *	0.08813	0.001	−0.4659	−0.1191
		Medium	−0.09675	0.06888	0.161	−0.2323	0.0388
**F3 Concerns**	Low	Medium	0.17533	0.11027	0.113	−0.0417	0.3923
		High	0.40723 *	0.10636	0.000	0.1979	0.6165
	Medium	Low	−0.17533	0.11027	0.113	−0.3923	0.0417
		High	0.23190 *	0.08313	0.006	0.0683	0.3955
	High	Low	−0.40723 *	0.10636	0.000	−0.6165	−0.1979
		Medium	−0.23190 *	0.08313	0.006	−0.3955	−0.0683

* The mean difference is significant at the 0.05 level.

**Table 25 behavsci-14-00563-t025:** ANOVA analysis of motivation to study the MDSE.

ANOVA					
Teacher Self-Efficacy Instructional Strategies					
	Sum of Squares	df	RMSE	F	Sig.
Between groups	13.340	4	3.335	3.021	0.018
Within groups	331.193	300	1.104		
Total	344.534	304			

**Table 26 behavsci-14-00563-t026:** Post hoc DMS analysis of motivation to study the MDSE.

Multiple Comparisons						
Dependent Variable: Teacher Self-Efficacy Instructional Strategies						
(I) Motivation	(J) Motivation	Mean Difference (I–J)	Std. Error	Sig.	95% Confidence Interval	
					Lower Bound	Upper Bound
**Vocation**	Possibility of finding a stable job	0.365 *	0.131	0.006	0.11	0.62
	For lack of a better option in my career path	0.597 *	0.295	0.044	0.02	1.18
	Influenced by a teacher who has influenced me in my education	0.418	0.278	0.134	−0.13	0.97
	Due to the influence of a family member who is or has been a teacher	0.543	0.278	0.052	0.00	1.09

* The mean difference is significant at the 0.05 level.

**Table 27 behavsci-14-00563-t027:** ANOVA analysis of CFDPAD dimensions.

ANOVA						
		Sum of Squares	df	RMSE	F	Sig.
**D1: “ Conditioning factors in the process of addressing diversity in the classroom”**	Between groups	55.660	21	2.650	2.597	0.000
	Within groups	288.874	283	1.021		
	Total	344.534	304			
**D2: “ Curricular and organizational response to diversity in the classroom”**	Between groups	42.615	23	1.853	1.724	0.023
	Within groups	301.918	281	1.074		
	Total	344.534	304			
**D3: “ Teacher training towards diversity”**	Between groups	57.813	27	2.141	2.069	0.002
	Within groups	286.720	277	1.035		
	Total	344.534	304			
**D5: “** **Teacher perception towards students with specific educational support needs”**	Between groups	27.702	10	2.770	2.571	0.005
	Within groups	316.832	294	1.078		
	Total	344.534	304			

**Table 28 behavsci-14-00563-t028:** Post hoc DMS analysis of CFDPAD dimensions.

Multiple Comparisons							
Dependent Variable	(I) Level of Teachers’ Self-Efficacy in Instructional Strategies	(J) Level of Teachers’ Self-Efficacy in Instructional Strategies	Mean Difference (I–J)	Std. Error	Sig.	95% Confidence Interval	
						Lower Bound	Upper Bound
**D1: “ Conditioning factors in the process of addressing diversity in the classroom”**	Low	Medium	−0.09305 *	0.04556	0.042	−0.1827	−0.0034
		High	−0.13423 *	0.04246	0.002	−0.2178	−0.0507
	Medium	Low	0.09305 *	0.04556	0.042	0.0034	0.1827
		High	−0.04118	0.03402	0.227	−0.1081	0.0258
	High	Low	0.13423 *	0.04246	0.002	0.0507	0.2178
		Medium	0.04118	0.03402	0.227	−0.0258	0.1081
**D2: “Curricular and organizational response to diversity in the classroom”**	Low	Medium	−0.02125	0.13226	0.872	−0.2815	0.2390
		High	−0.30144 *	0.12324	0.015	−0.5440	−0.0589
	Medium	Low	0.02125	0.13226	0.872	−0.2390	0.2815
		High	−0.28019 *	0.09875	0.005	−0.4745	−0.0859
	High	Low	0.30144 *	0.12324	0.015	0.0589	0.5440
		Medium	0.28019 *	0.09875	0.005	0.0859	0.4745

* The mean difference is significant at the 0.05 level.

**Table 29 behavsci-14-00563-t029:** ANOVA analysis of SACIE-R factors.

ANOVA						
		Sum of Squares	df	RMSE	F	Sig.
**F2 Feelings**	Between groups	39.226	8	4.903	4.754	0.000
	Within groups	305.308	296	1.031		
	Total	344.534	304			
**F3 Concerns**	Between groups	36.695	11	3.336	3.175	0.000
	Within groups	307.839	293	1.051		
	Total	344.534	304			

**Table 30 behavsci-14-00563-t030:** Post hoc DMS analysis of SACIE-R factors.

Multiple Comparisons							
Dependent Variable	(I) Level of Teachers’ Self-Efficacy in Instructional Strategies	(J) Level of Teachers’ Self-Efficacy in Instructional Strategies	Mean Difference (I–J)	Std. Error	Sig.	95% Confidence Interval	
						Lower Bound	Upper Bound
**F2 Feelings**	Low	Medium	0.09197	0.09492	0.333	−0.0948	0.2788
		High	0.19233 *	0.08845	0.030	0.0183	0.3664
	Medium	Low	−0.09197	0.09492	0.333	−0.2788	0.0948
		High	0.10036	0.07087	0.158	−0.0391	0.2398
	High	Low	−0.19233 *	0.08845	0.030	−0.3664	−0.0183
		Medium	−0.10036	0.07087	0.158	−0.2398	0.0391
**F3 Concerns**	Low	Medium	−0.15858	0.11480	0.168	−0.3845	0.0673
		High	0.11118	0.10698	0.299	−0.0993	0.3217
	Medium	Low	0.15858	0.11480	0.168	−0.0673	0.3845
		High	0.26976 *	0.08572	0.002	0.1011	0.4384
	High	Low	−0.11118	0.10698	0.299	−0.3217	0.0993
		Medium	−0.26976 *	0.08572	0.002	−0.4384	−0.1011

* The mean difference is significant at the 0.05 level.

**Table 31 behavsci-14-00563-t031:** ANOVA analysis of area of knowledge of higher studies of access to MDSE.

ANOVA					
Teacher Self-Efficacy Classroom Management					
	Sum of Squares	df	RMSE	F	Sig.
Between groups	19.024	4	4.756	3.203	0.013
Within groups	445.413	300	1.485		
Total	464.437	304			

**Table 32 behavsci-14-00563-t032:** Post hoc DMS analysis of area of knowledge of higher studies of access to MDSE.

Multiple Comparisons						
Dependent Variable: Teacher Self-Efficacy Classroom Management						
(I) Area of Knowledge	(J) Area of Knowledge	Mean Difference (I–J)	Std. Error	Sig.	95% Confidence Interval	
					Lower Bound	Upper Bound
**Social and Legal Sciences**	Science	0.247	0.136	0.071	−0.02	0.52
	Health Sciences	−0.097	0.288	0.736	−0.66	0.47
	Engineering and Architecture.	0.356 *	0.132	0.007	0.10	0.62
	Arts and Humanities	0.313 *	0.115	0.007	0.09	0.54

* The mean difference is significant at the 0.05 level.

**Table 33 behavsci-14-00563-t033:** ANOVA analysis of CFDPAD dimensions.

ANOVA						
		Sum of Squares	df	RMSE	F	Sig.
**D1: “Conditioning factors in the process of addressing diversity in the classroom”**	Between groups	78.086	21	3.718	2.724	0.000
	Within groups	386.351	283	1.365		
	Total	464.437	304			
**D2: “Curricular and organizational response to diversity in the classroom”**	Between groups	85.766	23	3.729	2.767	0.000
	Within groups	378.672	281	1.348		
	Total	464.437	304			
**D3: “Teacher training towards diversity”**	Between groups	74.593	27	2.763	1.963	0.004
	Within groups	389.844	277	1.407		
	Total	464.437	304			
**D4: “Teacher perception towards students with specific educational support needs”**	Between groups	42.559	14	3.040	2.090	0.012
	Within groups	421.878	290	1.455		
	Total	464.437	304			

**Table 34 behavsci-14-00563-t034:** Post hoc DMS analysis of CFDPAD dimensions.

Multiple Comparisons							
Dependent Variable	(I) Level of Teachers’ Self-Efficacy in Classroom Management	(J) Level of Teachers’ Self-Efficacy in Classroom Management	Mean Difference (I–J)	Std. Error	Sig.	95% Confidence Interval	
						Lower Bound	Upper Bound
**D1: “Conditioning factors in the process of addressing diversity in the classroom”**	Low	Medium	0.01083	0.03595	0.764	−0.0599	0.0816
		High	−0.06833	0.03872	0.079	−0.1445	0.0079
	Medium	Low	−0.01083	0.03595	0.764	−0.0816	0.0599
		High	−0.07916 *	0.03787	0.037	−0.1537	−0.0046
	High	Low	0.06833	0.03872	0.079	−0.0079	0.1445
		Medium	0.07916 *	0.03787	0.037	0.0046	0.1537
**D2: “Conditioning factors in the process of addressing diversity in the classroom”**	Low	Medium	−0.18142	0.10249	0.078	−0.3831	0.0203
		High	−0.45187 *	0.11039	0.000	−0.6691	−0.2346
	Medium	Low	0.18142	0.10249	0.078	−0.0203	0.3831
		High	−0.27045 *	0.10797	0.013	−0.4829	−0.0580
	High	Low	0.45187 *	0.11039	0.000	0.2346	0.6691
		Medium	0.27045 *	0.10797	0.013	0.0580	0.4829
**D4: “Formative teaching practice in addressing diversity”**	Low	Medium	−0.10214	0.06868	0.138	−0.2373	0.0330
		High	−0.27460 *	0.07398	0.000	−0.4202	−0.1290
	Medium	Low	0.10214	0.06868	0.138	−0.0330	0.2373
		High	−0.17246 *	0.07236	0.018	−0.3148	−0.0301
	High	Low	0.27460 *	0.07398	0.000	0.1290	0.4202
		Medium	0.17246 *	0.07236	0.018	0.0301	0.3148

* The mean difference is significant at the 0.05 level.

**Table 35 behavsci-14-00563-t035:** ANOVA analysis of SACIE-R factors.

ANOVA						
		Sum of Squares	df	RMSE	F	Sig.
**F1 Attitudes**	Between groups	43.475	14	3.105	2.139	0.010
	Within groups	420.962	290	1.452		
	Total	464.437	304			
**F2 Feelings**	Between groups	51.091	8	6.386	4.573	0.000
	Within groups	413.347	296	1.396		
	Total	464.437	304			
**F3 Concerns**	Between groups	72.119	11	6.556	4.896	0.000
	Within groups	392.318	293	1.339		
	Total	464.437	304			

**Table 36 behavsci-14-00563-t036:** Post hoc DMS analysis of SACIE-R factors.

Multiple Comparisons							
Dependent Variable	(I) Level of Teachers’ Self-Efficacy in Classroom Management	(J) Level of Teachers’ Self-Efficacy in Classroom Management	Mean Difference (I–J)	Std. Error	Sig.	95% Confidence Interval	
						Lower Bound	Upper Bound
**F2 Feelings**	Low	Medium	0.09543	0.07416	0.199	−0.0505	0.2414
		High	0.19887 *	0.07987	0.013	0.0417	0.3560
	Medium	Low	−0.09543	0.07416	0.199	−0.2414	0.0505
		High	0.10344	0.07812	0.186	−0.0503	0.2572
	High	Low	−0.19887 *	0.07987	0.013	−0.3560	−0.0417
		Medium	−0.10344	0.07812	0.186	−0.2572	0.0503
**F3 Concerns**	Low	Medium	0.11045	0.08794	0.210	−0.0626	0.2835
		High	0.44410 *	0.09472	0.000	0.2577	0.6305
	Medium	Low	−0.11045	0.08794	0.210	−0.2835	0.0626
		High	0.33365 *	0.09264	0.000	0.1513	0.5160
	High	Low	−0.44410 *	0.09472	0.000	−0.6305	−0.2577
		Medium	−0.3 3365 *	0.09264	0.000	−0.5160	−0.1513

* The mean difference is significant at the 0.05 level.

## Data Availability

Data available on request from the authors.

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
