# Peer review of "Analysis of Teacher Self-Efficacy and Its Impact on Sustainable Well-Being at Work"

_behavsci, 2024, doi:10.3390/bs14070563_

Round 1
Reviewer 1 Report
Comments and Suggestions for Authors
Thank you for the opportunity to review the manuscript, Analysis of teacher self-efficacy and its impact on Sustainable Well-being at Work. The authors examine self-efficacy in various areas and comparing demographic information to levels of self- efficacy, specifically in inclusion, diversity, and classroom management domains. The authors write clearly but there are areas that need addressed.
Title: The title of the paper does not align neatly with the manuscript’s purpose. The findings and discussion in the paper focus around self-efficacy in several areas and related to demographic variances. Where is the section that addresses how the findings show the “impact on sustainable well-being at work”?
Literature Review:
The bulk of the literature focuses on work around self-efficacy. What research exists that specifically focuses on teacher self-efficacy regarding inclusion, diversity, and classroom management? Is there research regarding self-efficacy and a Master’s level program? Or are you identifying a gap? What about SE differences between men and women or those in private v. public universities? This is unclear and needs to be addressed.
Methods and Materials:
Various instruments are discussed and used in this study. In the reporting of the findings, there are references to items, yet the reader has no indication of what the items are without delving into the literature for each of the various instruments. An appendix with items focused on for this study would aid in understanding of how the instruments were used.
Results:
There are a variety of statistical analyses that were run (ANOVA, t-test, correlation, etc.) yet there are not corresponding tables that depict those results. This makes it cumbersome for readers to digest as they only have the written text version of these findings.
Table 3: The questions listed are open-ended. How were these answered by participants to generate means?
Page 17 = t-test results – no table
Page 18: Correlation discussed – no correlation table with the details. The one provided only shows total SE but authors discuss the low, medium, and high SE. Either a more clear correlation chart is needed or more explanation on how you determined findings based on efficacy level.
Anova table?? One is needed
Top of page 8 = You reference Dimension 2 had the strongest correlation but the chart Table 5 does not agree with that statement.
There are various other places where results are not reported in tables and this should be added.
Discussion
Authors have some paragraphs that are one sentence in length. More depth should be added to these paragraphs. The authors also do not align their discussion with what they know from the literature. Do your findings show something new and novel or are there areas where your research supports current research? What implications does this have for supporting teachers in inclusive practices, diversity, and classroom management?
Author Response
Thank you for the opportunity to review the manuscript, Analysis of teacher self-efficacy and its impact on Sustainable Well-being at Work. The authors examine self-efficacy in various areas and comparing demographic information to levels of self- efficacy, specifically in inclusion, diversity, and classroom management domains. The authors write clearly but there are areas that need addressed.
Title: The title of the paper does not align neatly with the manuscript’s purpose. The findings and discussion in the paper focus around self-efficacy in several areas and related to demographic variances. Where is the section that addresses how the findings show the “impact on sustainable well-being at work”?
Response: Thank you very much for pointing it out. This has been changed in the manuscript:
These heightened feelings of self-efficacy, positively linked to instructional effectiveness, prompt teachers to engage in organizational planning and demonstrate a proactive approach towards the challenges and obstacles encountered in their daily teaching. Furthermore, teacher self-efficacy can mitigate and alleviate acute stressors associated with work, thereby enhancing teachers' overall well-being. [50]
And these results are very important because the examination of teacher self-efficacy and its implications for sustainable workplace well-being highlights its pivotal role in shaping educators' perceptions of their profession, classroom dynamics, and their influence on student learning outcomes. Educators with heightened self-efficacy are inclined to adopt effective pedagogical strategies and techniques, leading to heightened student engagement and academic success. Furthermore, teacher self-efficacy directly impacts performance, reflecting overall competence [58].

Reviewer 2 Report
Comments and Suggestions for Authors
Congratulations on an excellent paper. The only suggestion I would make would be one related to implications for the profession. Your findings are not new but again emphasise the quality of training that should be built on when the pre-service teachers enter the profession as a beginning teacher. There should be a mentoring program to ensure that the beginning teacher develops contextual skills, develop reflection skills and as such learn from their practice - especially in the first few years of their teaching.
Author Response
Congratulations on an excellent paper. The only suggestion I would make would be one related to implications for the profession. Your findings are not new but again emphasize the quality of training that should be built on when the pre-service teachers enter the profession as a beginning teacher. There should be a mentoring program to ensure that the beginning teacher develops contextual skills, develop reflection skills and as such learn from their practice - especially in the first few years of their teaching.
Response: Thank you very much for pointing it out. This has been changed in the manuscript:
There should be a mentoring program to ensure that the beginning teacher develops contextual skills, develop reflection skills and as such learn from their practice - especially in the first few years of their teaching

Reviewer 3 Report
Comments and Suggestions for Authors
Dear authors,
The topic of self-efficacy of the teachers among students is a good topic that brings original input to the topic of educational field. The theory is not structured on hypotheses. It is very important that the main hypotheses to be presented in the theoretical framework.
The methodology is well structured around correlation statistics results, but the hypotheses formulated around the tested relations of self-efficacy and attitudes, feelings and concerns.
Discussions should better emphasize the theoretical contribution of the paper, as the practical contributions are very well stated.
Comments on the Quality of English LanguageThe text is very clear, the quality of English language is good.
Author Response
Review 3
Dear authors,
The topic of self-efficacy of the teachers among students is a good topic that brings original input to the topic of educational field. The theory is not structured on hypotheses. It is very important that the main hypotheses to be presented in the theoretical framework.
The methodology is well structured around correlation statistics results, but the hypotheses formulated around the tested relations of self-efficacy and attitudes, feelings and concerns.
Discussions should better emphasize the theoretical contribution of the paper, as the practical contributions are very well stated.
Response: Thank you very much for pointing it out. This has been changed in the manuscript:
We hypothesize that a heightened sense of teacher self-efficacy will be related to a more favorable disposition towards diversity, fewer related concerns, and a higher level of knowledge and competencies acquired in the MDSE program for their development as inclusive teachers in a school for all.
Round 2
Reviewer 1 Report
Comments and Suggestions for Authors
Thank you for your resubmission and edits. Many of the edits addressed my concerns.
Please see the attached feedback. Some of the original feedback was not addressed. I tried to be concisely express the areas of concern.

Author Response
Thank you very much, your comments have greatly improved the quality of the document.
Thank you sincerely.

Round 3
Reviewer 1 Report
Comments and Suggestions for Authors Methods- Crosstabulations (starting line 226) - Using chi-squared often results in generous p-values. Please explain how you ensured the use of a more rigorous p-value or justify why you used the p-value you did. Discussion - first paragraph is one long sentence. This has been a consistent pattern in this manuscript. Paragraphs must be developed and not consist of a singular sentence. I recommend adding something like "As such, the current research contributes to a gap in literature around...." You might even add another sentence about who you hope will benefit from this new -found knowledge.
Author Response
Again thank you very much for your suggestions, as always, a success.
